

# Global physics-based database of injection-induced seismicity

Iman R. Kivi[1,2,3], Auregan Boyet[1,2,3], Haiqing Wu[3,4], Linus Walter[1,2,3], Sara Hanson-Hedgecock[1,2,3], Francesco Parisio[1,2,3], Victor Vilarrasa[1,3]

[1]Global Change Research Group (GCRG), IMEDEA, CSIC-UIB, Esporles, Spain

[2]Institute of Environmental Assessment and Water Research, Spanish National Research Council (IDAEA-CSIC), Barcelona, Spain

[3]Associated Unit: Hydrogeology Group (UPC-CSIC), Barcelona, Spain

[4]Department of Civil and Environmental Engineering (DECA), Universitat Politécnica de Catalunya (UPC), Barcelona, Spain

*Correspondence to*: Iman R. Kivi (iman.rahimzadeh@idaea.csic.es)

**Abstract.** Fluid injection into geological formations for energy resource development frequently induces (micro)seismicity. If intensely shaking the ground, induced earthquakes may cause injuries and/or economic loss, with the consequence of jeopardizing the operation and future development of these geoenergy projects. To achieve an improved understanding of the causes of induced seismicity, develop forecasting tools, and manage the associated risks, a careful examination of seismic data

from reported cases of induced seismicity and the parameters controlling them is necessary. However, these data are hardly gathered together and are time-consuming to collate as they come from different disciplines and sources. Here, we present a publicly available, multi-physical database of injection-induced seismicity (Kivi et al., 2022a; https://doi.org/10.20350/digitalCSIC/14813), sourced from an extensive review of published documents. Currently, it contains 158 datasets of induced seismicity driven by various subsurface energy-related applications worldwide. Each dataset covers a

wide range of variables, delineating general site information, host rock properties, in situ geologic and tectonic conditions, fault characteristics, conducted field operations, and recorded seismic activities. We publish the database in flat-file formats (i.e., .xls and .csv tables) to facilitate its dissemination and utilization by geoscientists while keeping it directly readable by computer codes for convenient data manipulation. The multi-disciplinary content of this database adds unique value to databases focusing only on seismicity data. In particular, the collected data aims at facilitating the understanding of the

spatiotemporal occurrence of induced earthquakes, the diagnosing of potential triggering mechanisms, and the developing of scaling relations of maximum possible earthquake magnitudes and operational parameters. Conclusively, the database will boost research in seismic hazard forecasting and mitigation, paving the way for increasing contributions of geoenergy resources to meeting net-zero carbon emissions.



## 1 Introduction

Fluid injection into and withdrawal from the subsurface, deep underground mining, and reservoir impoundment are some of the most prominent causes of induced seismicity, which became a global problem in the past decade (see Foulger et al., 2018 for a comprehensive review). In this period, the rate of induced earthquakes with magnitudes M>3 has grown threefold in western Canada (Atkinson et al., 2020) and tenfold in Oklahoma (Ellsworth, 2013). While the increased levels of seismicity in western Canada are broadly attributed to the hydraulic fracturing of ultralow-permeability shales to commercially exploit

unconventional oil and gas (Bao and Eaton, 2016), seismic activity in the midwestern United States has increased principally as a result of large-volume wastewater disposal in deep formations (Shirzaei et al., 2016). Several large earthquakes have also been triggered in the course of geothermal energy exploitation in response to fluid injection, extraction, circulation, and, more importantly, hydraulic stimulation of naturally low-permeability hot formations to develop Enhanced Geothermal Systems (EGS, Evans et al., 2012; Buijze et al., 2019). Earthquakes induced by these geoenergy activities were occasionally felt by the

local population and even resulted in injuries to people, and damage to buildings and infrastructure, causing early termination of projects and loss of investment (Häring et al., 2008; Cesca et al., 2014; Lee et al., 2019). Utilization of the subsurface for energy purposes is likely to intensify in the upcoming decades, mainly driven by applications connected to the energy transition, such as geologic Carbon Capture and Storage (CCS) (Ringrose et al., 2021) and geologic hydrogen storage (Heinemann et al., 2021). Injection-induced seismicity remains one of the largest liabilities of geoenergy projects and can

potentially have a vast societal, environmental, and economic impact (Verdon, 2014; Vilarrasa et al., 2019). Therefore, minimizing the risks associated with induced earthquakes is a prerequisite for the secure and sustainable deployment of geoenergy applications worldwide (see Fig. 1 for an overview of geoenergy projects triggering seismicity).







**Figure 1. Schematic illustration of geoenergy applications linked with induced seismicity. Earthquakes have reportedly been induced by tight and shale gas fracturing, conventional oil and gas development activities, deep wastewater disposal, geologic storage of natural gas or CO₂, geothermal energy exploitation and research projects.**

The recent surge in the number of injection-induced earthquakes has drawn considerable attention in the seismological and hydrogeological research communities. Scientific efforts are mainly focused on understanding the triggering mechanisms of induced earthquakes, forecasting the seismic risk/hazards and developing mitigation/management strategies. There are a variety of approaches to seismic hazard forecasting and management, which are commonly categorized into probability- and physics-based approaches depending on the utilized input data, applied processing methods and outcomes.



Probability-based techniques, independent of the physics that induce the earthquakes, strive to develop a quasi-real-time

prediction of the seismicity rate and magnitude evolution. To this end, these approaches inherit two fundamental laws from statistical seismology (Ogata, 1988; Shapiro et al., 2010; Bachmann et al., 2011): the Gutenberg–Richter (G-R) law (Gutenberg and Richter, 1942), which describes the frequency-magnitude distribution of earthquakes, and the Omori-Utsu law (Utsu, 1961), delineating the aftershock decay. Both statistical approaches are based on model calibration against catalogs of monitored induced seismicity. However, the statistics evolve with fluid injection, presenting more frequent small earthquakes

during injection and more larger earthquakes after the stop of injection (e.g., Ruiz-Barajas et al., 2017). Such evolution trends of the earthquake magnitudes limit the predictive capability of probability-based methods.

Physics-based approaches aim at constraining the spatiotemporal evolution of seismicity by considering the underlying triggering mechanisms. A fault reactivates when the shear stress acting on the fault plane exceeds its frictional strength (Jaeger et al., 2009). Accordingly, stress perturbations and/or strength alterations on seismogenic faults (faults prone to seismic slip),

driven by coupled thermal-hydraulic-mechanical-chemical (THMC) processes of fluid flow in porous and/or fractured rocks, may result in earthquakes. The impacts of THMC processes on induced seismicity have been largely acknowledged in recent years mainly by incorporating them into process-based modeling of induced earthquakes (Cappa and Rutqvist, 2011; Ghassemi and Zhou, 2011; De Simone et al., 2017; Vilarrasa et al., 2021; Kivi et al., 2022b). Nevertheless, the governing mechanisms of seismic sequences (1) unexpectedly triggered after the causative operation ceased, i.e., post-injection seismicity (Segall and

Lu, 2015; Johann et al., 2016), (2) located tens of kilometers away from the operation sites (Goebel et al., 2017; Yeck et al., 2017) or (3) vertically offset by up to several kilometers from the fluid injection/withdrawal horizon (Eyre et al., 2019; Vilarrasa et al., 2021; Zhai et al., 2021) remain largely elusive. Using physically-sound models with varying degrees of sophistication to reproduce seismicity recorded in case examples around the world give invaluable insights into the causes of these challenging seismic sequences. Indeed, such studies should enable advance on two fronts: shedding light on earthquake-

triggering mechanisms and developing a proactive framework for future seismic hazard quantification and management. However, these modeling efforts rely on access to several parameters, including geological setting, multi-physical reservoir rock and fault properties, in situ stress, pressure and temperature distributions across the reservoir and details of the conducted industrial operations and recorded seismicity. The main limitation is that these variables are scattered across multiple disciplines and hardly gathered together in reported sites of induced seismicity.

Considerable efforts have been devoted to predicting or drawing bound on the maximum earthquake magnitude, $M_{max}$, by scaling it with (1) the cumulative injected fluid volume (McGarr, 2014; Galis et al., 2017), (2) the initial stress state (Li et al., 2021), (3) the number of induced earthquakes (van der Elst et al., 2016), (4) dimensions of the stimulated volume (Shapiro et al., 2011), or (5) the elapsed time from the onset of injection to the earthquake occurrence (Shapiro et al., 2021). However, caution should be taken when employing scaling relations of induced seismicity. For example, the 2017 $M_{max}$ 5.5 Pohang

earthquake in Korea, triggered by stimulation of an EGS, is a well-known outlier in the magnitude scaling relations, where the injected fluid volume was 500 times smaller than the amount expected to induce the earthquake (Lee et al., 2019). Data



emerging with the growing incidences of induced earthquakes present an unprecedented opportunity to verify the reliability of the existing seismic forecasting models and develop alternatives better indicative of the underlying physics.

The many review articles and reports scrutinizing induced seismicity (Suckale, 2009; Evans et al., 2012; National Research
Council, 2013; Gaucher et al., 2015; Grigoli et al., 2017; Keranen and Weingarten, 2018; Vilarrasa et al., 2019; Ge and Saar, 2022) converge on a common conclusion: a comprehensive and publicly accessible database of seismicity and parameters controlling it from historical cases would be of utmost value to improved characterization of induced seismicity and informed management of its risks. Wilson et al. (2017) and Foulger et al. (2018) presented an exhaustive inventory of all (potentially) induced earthquakes called the HiQuake database, with the most recent updates being available in an online repository (The
Human-Induced Earthquake Database (HiQuake): https://inducedearthquakes.org/, last access: 3 June 2022). Nevertheless, the covered data by HiQuake is primarily restricted to seismicity catalogs and relatively few operational parameters, while key tectonic, rock and fault properties are missing; filling the gap is the main goal of this study.

We have developed a multi-physical database of injection-induced seismicity in the framework of the ERC-funded project GEoREST (predictinG EaRthquakES induced by fluid injecTion, grant agreement No. 801809). The database gathers a
publicly accessible compilation of parameters that control injection-induced seismicity and are relevant to geoenergy developments. Here, we provide an overview of the database content and structure, present the resources and the criteria considered for the collection and curation of data and formally release the current state of the database as its first version in flat-file formats (Kivi et al., 2022a; https://doi.org/10.20350/digitalCSIC/14813). In total, 71 parameters, categorized into 7 disciplines, have been collected for 158 cases of induced seismicity. The database will be updated in the future for new cases
of induced earthquakes and already missing case histories, particularly those from the petroleum industry, if data become available. The large number of case examples and diversity of input parameters make the collected datasets very well suited for testing new scaling relations for maximum earthquake magnitude forecasting.

## 2 Description of the database content

### 2.1 Database structure

The database is licensed under Creative Commons CC BY 4.0 International License and is publicly accessible. The compilation contains 158 notable cases of injection-induced seismicity together with multi-physical parameters characterizing the seismic events. It practices FAIR guiding principles for data management that assists in making the database Findable, Accessible, Interoperable and Reusable by humans and machines (Wilkinson et al., 2016). We provide the database in two flat-file formats: the first as a single Microsoft Excel spreadsheet to keep it as simple and friendly as possible to researchers and inexperienced
end-users, and the second as a .csv file, representing a standard machine-readable format for direct implementation of data in model developments.

We build our database mainly upon HiQuake, the holistic and invaluable compilation of earthquakes proposed to be induced by human activities. However, judging if the earthquakes were definitely human-induced is sometimes challenging and



subjected to inevitably varying opinions among researchers. These uncertainties mainly grow when discriminating between

natural and induced earthquakes if located at seismically active plate boundaries. We rely on HiQuake in the consensus that anthropogenic activities induced all reported earthquakes. Thus, we neither independently assess nor negate the induced or natural essence of these cases. Our database puts exclusive emphasis on injection-induced earthquakes. Our attempts to develop this collection consist in (1) complementing data entries for general site characteristics, operational parameters, and seismicity data to which HiQuake has been restricted, and (2) collecting data for 41 additional input parameters concerning reservoir rock

properties, fault characteristics, and in situ stress and pressure data through comprehensively reviewing near 500 scientific resources. This extensive set of input parameters is necessary to achieve a mechanistic understanding of induced seismicity and develop forecasting models.

A fundamental criterion for sites to be included in the database is data availability on a publicly accessible scientific basis (Sect. 2.2.9). Thus, we do not list in the database several cases associated, in particular, with conventional hydrocarbon

development projects that lack information about the hydrogeological and geomechanical properties, potentially due to confidentiality (see Sect. 2.3 for more details of data curation). We categorize the remaining injection-induced earthquakes based on the geoenergy application types into (1) geologic gas storage (including both natural gas and $CO_2$ storage), (2) geothermal energy exploitation, (3) tight and shale gas fracturing, (4) research projects, and (5) wastewater disposal, all playing prominent roles in sustainable and green energy transition (IPCC, 2018).

Every case of induced seismicity belonging to an individual project or to separate phases of a project is listed in a distinct row. The latter case scenarios are particularly relevant to the circulation, injection, or stimulation phases of the same geothermal plant (e.g., the Cooper, Insheim or Soultz geothermal sites) and multistage hydraulic fracturing of shale gas resources from a single well pad (e.g., Fox Creek sequences). In contrast, only one sequence possessing the largest maximum magnitude is considered for long-lived injection operations, like the Geysers EGS project, which presumably result in multiple distinct

seismic sequences, and for repeating scientific injections at centimeter- or decameter-scale rock laboratories. The collected cases are sorted alphabetically with reference to the project type, then to the country, and finally to the site location. The users can simply adapt the sorting to any order of interest.

The parameters are listed in separate columns and structured into seven main sub-tables, succeeded by two single entities of complementary remarks and references, to increase readability and facilitate data usage (see Fig. 2 for a complete list of

database parameters). The first row of the database contains the sub-table headings, which respectively comprise a series of interconnected entries of project information, reservoir rock properties, in situ tectonic and pressure data of the site, fault characteristics, injection data, general seismic records, and the maximum magnitude event information. A total of 71 individual input parameters delineate each case of induced earthquake and are labeled with their units in the second row. Subdivisions may apply to some parameters, essentially to present a range instead of a single value. The aforementioned three levels of

parameter definition are merged into a unified, short and self-explanatory naming convention in the format of "data type_parameter name_subdivisions" on the fourth row. For example, "fault_dens_min" points to the minimum value of the density of the rock forming the fault [kg/m$^3$]. The adopted naming convention enables rendering the database in separate





single-header and uniquely described columns to make it machine-readable and, thus, easy to process by other researchers. The database is accompanied by a dictionary that maps the abbreviated names of all parameters to their full meaning. A detailed 160 explanation of all entities, divided into different data types, is documented in Sect. 2.2.

---

### *Database variables*

**General project information**
- Case number
- Country
- Location
- Latitude (°)
- Longitude (°)
- Project type
- Sub-class

**Host rock properties**
- Formation name
- Stratigraphy
- Fracture density (count/m)
- Density (kg/m³)
- Porosity (-)
- Permeability (m²)
- Young´s modulus (GPa)
- Poisson´s ratio (-)
- Bulk modulus (GPa)
- Shear modulus (GPa)
- Biot coefficient (-)
- Friction angle (°)
- Cohesion (MPa)
- UCS (MPa)
- Tensile strength, $T_0$ (MPa)
- Thermal Conductivity (W/mK)
- Thermal expansion coefficient (1/°K)

**Site characteristics**
- Depth of basement (m)
- Stress
- Overburden stress, $\sigma_v$ (MPa)
- Min. horizontal stress, $\sigma_h$ (MPa)
- Max. horizontal stress, $\sigma_H$ (MPa)
- Max. horizontal stress direction (°)
- Pore pressure (MPa)
- Temperature (°C)

**Fault properties**
- Strike (°)
- Dip (°)
- Dip direction (°)
- Fault name
- Fault type
- Thickness (m)
- Core thickness (m)
- Distance from injection (m)
- Intersection depth (m)
- Density (kg/m³)
- Porosity (-)
- Permeability (m²)
- Normal Stiffness (GPa/m)
- Shear stiffness (GPa/m)
- Dilation angle (°)
- Young´s modulus (GPa)
- Poisson´s ratio (-)
- Friction angle (°)

**Injection data**
- Depth of injection (m)
- Injection type
- Injection start date
- Fluid type
- Injection temperature (°C)
- Max. Injection rate (m³/s)
- Injected volume (m³)
- Net injection volume (m³)
- Max. wellhead pressure (MPa)
- Max. bottomhole pressure (MPa)

**Induced seismicity information**
- Seismicity onset
- Time between injection and first seismicity (d)
- Number of events
- Depth of seismicity (m)
- G-R law parameters, before injection
- G-R law parameters, during injection
- G-R law parameters, after injection

**Seismicity information for $M_{max}$**
- $M_{max}$
- Type of $M_{max}$
- Depth of $M_{max}$ (m)
- Distance from injection (m)
- Date of $M_{max}$

**Complementary remarks**

**References**

**Figure 2. Schematic structure of the database representing different sub-tables and the associated properties.**

---

### 2.2 Input parameters

We describe in the following parameters included in our database. In the context of physics- or statistic-based approaches, supported by field observations on availability, we argue how different parameters are relevant to the improved understanding





and forecasting of induced seismicity. We also comment on the availability of data, common approaches to measure (or record) the data and the way we report them.

### 2.2.1 General project information

This sub-table comprises all data relevant to the type and location of the project that led to the respective induced earthquake. The first column contains the project number. We subsequently list the country, name, and coordinates (i.e., latitude and longitude in the WGS84 reference system in decimal degrees) of the place or site where the project was conducted. The project name distinguishes between separate phases of a project leading to independent seismic sequences on the spatial and temporal domains. We complement project information by documenting both the project category and sub-category.

The sub-category is of particular importance to induced seismicity in geothermal reservoirs in which the rate and the total net volume of injected fluid vary widely among different sub-categories. Various operations during geothermal energy exploitation have reportedly been associated with induced seismicity, offering corresponding sub-categories for this project type: drilling, preliminary hydraulic test, circulation, injection, and enhanced geothermal systems (EGS). While the produced and injected fluid masses during circulation are commonly balanced, huge amounts of fluid at elevated rates and pressures are injected during the stimulation phase of EGS projects (Evans et al., 2012). The sub-categories also differentiate between research projects injecting fluid into centimeter-scale specimens, decameter-scale Underground Rock Laboratories (URLs), and deep boreholes.

We gather together natural gas storage and CCS projects under the broader framework of geologic gas storage because they may share common operational characteristics, physical properties of the injected fluid and seismicity-triggering mechanisms. Arguably, megatonne geologic carbon storage projects have remained seismically quiescent (Vilarrasa and Carrera, 2015; Ge and Saar, 2022). However, the need for rapid and massive CCS scale-up raises concerns about basin-wide pore pressure buildups and, thus, the likelihood of triggering induced earthquakes (Zoback and Gorelick, 2012; Verdon, 2014). These concerns are consistent with the notable case of the 2013 moment magnitude ($M_w$) 4.2 earthquake at Castor, Spain, linked to underground natural gas storage (Vilarrasa et al., 2021) and encourage revisiting in more detail the issue of induced seismicity during large-scale underground gas storage.

### 2.2.2 Reservoir rock properties

This sub-table provides data on the hydrogeological properties of the host rock that may affect induced seismicity calculations. We include the target formation name and the stratigraphy in the first and second columns of this sub-section, respectively. Information about the formation name facilitates cross-linking rock properties in different projects targeting the same formation. Such correlations help arrive at rough estimates of rock properties where direct measurements are missing (see Sect. 2.3). We provide 15 different rock parameters representing physical, hydraulic, poroelastic, thermal, and failure characteristics of the formations (see Fig. 2 for a detailed list of these parameters). In particular, we include elastic moduli, Biot coefficient,



porosity, and permeability of the rock, which are crucial to assessing triggering mechanisms and forecasting the timing and

magnitude of induced earthquakes, as briefly described below.

The temporal and spatial evolution of pore pressure in porous media, rendering a basic mechanism for inducing seismicity, is controlled by hydraulic diffusivity $D = k/\mu S$, in which $k$, $\mu$ and $S$ denote the rock permeability, dynamic fluid viscosity and storage coefficient, respectively (Rice and Cleary, 1976). Although the storage coefficient takes different formulations depending on the applied loading conditions, it primarily depends on the rock porosity, $\varphi$, and bulk moduli of the fluid, $K_f$,

solid grains, $K_s$, and rock skeleton under drained conditions, $K$ (Cheng, 2016). Pore pressure changes give rise to poroelastic stresses whose magnitudes depend on the rock stiffness (elastic moduli) and the Biot effective stress coefficient, defined as $\alpha = 1 - K/K_s$. Besides, cooling effects driven by long-term cold fluid injection (particularly during geologic carbon storage and geothermal energy exploitation) result in thermoelastic stresses, proportional to the rock stiffness and linear rock expansion coefficient, $\alpha_T$ (De Simone et al., 2017). Accordingly, McGarr (2014) has argued that the stiffer the rock, the larger the pore

pressure enhancement resulting from a constant volume of injected fluid and thus, the larger the shear stress buildup on the fault and the maximum expected earthquake magnitude.

The main data source for this class of parameters is commonly laboratory measurements on rock specimens retrieved preferably from depth, otherwise from outcrops, but also wireline logging interpretations and field tests. It should be borne in mind that field tests may better represent the in situ rock mass behavior, particularly if it is intensively fractured. The measured values

may significantly differ from laboratory inspections of nominally intact specimens, featuring differences in intrinsic permeabilities of several orders of magnitudes in fractured crystalline and argillaceous rocks (Brace, 1980). To account for the inherent heterogeneity of the rock and measurement uncertainties (see Sect. 2.3 for further details), we give the minimum and maximum values of the collected data pool. Laboratory measurements of rock porosity, permeability, and elastic moduli are more common and may span a wide range. Therefore, we also report for these parameters the average value, dealing partly

with the statistical distribution of the measured data. In addition, if the measurements only contain a single data point, particularly for values deduced from field tests, it is given as the average value across the injection interval.

### 2.2.3 In situ geologic and tectonic characteristics

In this section, we primarily verify the approximate depth of the crystalline basement where the crust is widely accepted to be critically stressed (Townend and Zoback, 2000) and the strength properties of faults render them more susceptible to seismic

slip (seismogenic faults, Verberne et al., 2020). These characteristics of the crystalline basement are consistent with observations of the nucleation of the vast majority of large earthquakes in the basement (Horton, 2012; Goebel et al., 2017; Buijze et al., 2019; Williams-Stroud et al., 2020). A notable example comes from wastewater disposal in Oklahoma where the released seismic moment was found to be strongly correlated with injection depth relative to the crystalline basement (Verdon, 2014; Hincks et al., 2018). We also describe the present-day in situ stress magnitudes, directions, and regime, as well as the

reservoir pressure and temperature. These factors play a first-order role in earthquake rupture and coupled THMC processes



that control the evolution of seismic hazards in time and space. The influence of the regional stress regime on fault stability changes induced by poroelastic and thermoelastic stresses has been well acknowledged through numerical simulations (Vilarrasa, 2016; Fan et al., 2019).

The magnitudes of the stress components and pressure are assumed to follow linear relationships with depth, characterized by

gradient "m" and surface value "n". The linear trends are commonly forced to pass through the origin, which entails n=0 (zero value at the surface). The linearity assumption is routinely adopted to evaluate in situ stress and pressure profiles and is deemed valid for relatively homogeneous, short depth intervals. We calculate, based on the established linear fittings, the maximum and minimum values of pressure and stress components corresponding to the bottom and top of the injection interval, respectively. Thus, missing either of the two values means that the injection depth interval is not constrained on one side.

The overburden stress and pore pressure gradients are inferred in a straightforward manner from density logs and well-test data, respectively. On the contrary, a range of wellbore measurements and techniques should be combined to delineate in situ horizontal stresses: (1) caliper or imagery detection of breakouts and drilling-induced tensile fractures (DITF) from which horizontal stress directions can be deduced, (2) leak-off or minifrac tests to estimate the minimum horizontal stress magnitude and finally (3) theoretical replication of the breakout and DITFs occurrences within the crustal strength bounds to constrain

the maximum horizontal stress magnitude (Haimson and Cornet, 2003; Zoback et al., 2003). The stress magnitude and regime determined from this integrated approach are widely found to correlate well with focal mechanisms of induced earthquakes, for instance, in Fox Creek (Shen et al., 2019) or Basel (Valley and Evans, 2019). If data from these sources are not available, we do not rely on focal plane solutions of natural earthquakes, accessible from the well-documented world stress map (Heidbach et al., 2018). The reason is that natural earthquakes likely belong to the deep crust where stress conditions do not

necessarily coincide with injection depths.

### 2.2.4 Fault properties

We collect a wide range of fault parameters including the name, type (normal, strike-slip, and reverse or a combination), orientation (strike and dip), total thickness, core thickness, location with respect to injection, and hydraulic and mechanical characteristics (Fig. 2 presents the full list of parameters). Knowing the fault name, valuable additional information may be

inferred from the existing databases of fault properties (e.g., Scibek, 2020 provide a worldwide database of fault permeability). The considered parameters in the database are essential to assess the slipping tendency of the fault. The pore pressure and stress distribution along the faults, originating from remote injection source(s), are usually calculated numerically. The numerical models represent the fault either as approximately planar discontinuities or as equivalent continuum porous media (Cappa and Rutqvist, 2011; Berre et al., 2019). While the former approach treats the fault explicitly using its normal and shear

stiffnesses, the latter needs two independent elastic moduli describing the deformation of fault-forming material. Thus, we list both sets of parameters although they can be approximated from each other by knowing the density of fractures or planes of weakness along the fault strike (Zareidarmiyan et al., 2020). We document the static friction and dilation angles respectively



as measures of the intrinsic resistance of the fault against slip initiation and the fault´s tendency to dilate as a result of slipping. From the hydraulic point of view, faults can act as barriers or conduits to flow along and across them (Caine et al., 1996), the

choice of which may strongly impact fault stability (Vilarrasa et al., 2016; Wu et al., 2021; Kivi et al., 2022b). The fault architecture may be extremely complex, producing anisotropic and heterogeneous permeability fields (Rinaldi et al., 2014). However, given the data scarcity, we do not account for such complexities and use single-value (scalar) hydromechanical parameters to represent isotropic and homogeneous faults.

The fault orientation and slip types are primarily derived from the earthquake focal mechanisms. The hydraulic permeability,

stiffness, and frictional strength of faults can be directly measured from laboratory tests on representative outcrop samples or retrieved cores from depth. These parameters can also be determined from appropriately designed and monitored injection experiments either at underground rock laboratories or field scales. Assuming that pore pressure perturbations (diffusion-like process) stand solely as the triggering mechanism for induced earthquakes, observations of the spatio-temporal migration of seismic events may provide valuable estimates of the average fault permeability (Shapiro et al., 1997; Talwani et al., 2003).

Arguably, the inferred values pose an upper bound limit to the possible range of fault permeability.

### 2.2.5 Injection data

We document the operational parameters that are of paramount importance in understanding and predicting induced earthquakes, as well as mitigation of seismic risks (Ge and Saar, 2022). Data include the injection depth interval together with the start date and specific remarks on the injection. These remarks mainly concern fluid injection protocols (commonly

constant-rate, stepwise-rate increase, and cyclic schemes) and applications not already mentioned in the project category or sub-category lists (e.g., pre-injection tests, the main injection stages or reinjections). We also list the fluid type, injection temperature, and viscosity if explicitly reported in the literature. Otherwise, one can estimate the viscosity from the fluid type and the injection temperature using appropriate equations of state. Besides, the temperature difference between the injected fluid and the reservoir generates thermal stresses that may control the stability of adjacent (Parisio et al., 2019) and even distant

faults (Kivi et al., 2022b). Although these thermal effects are more pronounced in geothermal systems, unambiguously due to elevated differential temperatures, non-negligible impacts are anticipated during geologic carbon storage (Vilarrasa and Rutqvist, 2017).

We gather together information on the cumulative volume of injected fluid and the maximum injection pressure and rate. These parameters primarily control the disturbance of pore pressure and stress in the subsurface and, thus, the possibility of inducing

earthquakes as described in the following. McGarr (2014) pioneered a relationship between the maximum anticipated earthquake magnitude and the injected fluid volume, turning to a popular and widely-cited approach to deal with the injection-induced seismicity risk. Besides, induced seismicity observed early during the stimulation phase and adjacent to the wellbore in a number of geothermal reservoirs has been closely correlated with high injection pressures (Zang et al., 2014). In contrast, adopting a poroelastic model of earthquake nucleation, Alghannam and Juanes (2020) argued that the likelihood of triggering





seismicity depends strongly on the injection rate rather than the magnitude of generated overpressure. In this sense, for a given total volume of injected fluid, the faster and larger the injection rate increase, the more frequent the seismicity. Statistical analyses also show that the dramatic growth of seismic activities in the central and eastern USA that began in 2009 is more likely linked to high-rate water disposal wells than low-rate wells (Weingarten et al., 2015, Langenbruch and Zoback, 2016). These analyses suggest that the fluid injection rate, among other operational parameters, may pose a first-order control on

seismicity risk. Ongoing research into understanding the interplay between the mentioned operational parameters may help come up with novel strategies to forecast and mitigate induced seismicity.

We discriminate between the total and net fluid volumes injected up to the time of the maximum magnitude earthquake (the latter is simply defined as the injected minus produced fluid volumes). The net injection volume for fluid circulation during geothermal energy exploitation may or may not equal zero, depending on the (im)balance between injection and production

rates. Missing records of either of the two rates would lead to high uncertainty in estimating the net injection volume in long-term circulation systems. Furthermore, we consider two measures for the injection pressure: wellhead pressure and bottom hole pressure. The geothermal gradient and friction of the working fluid generated in the annulus may impose non-trivial impacts on the bottom hole pressure (Pan and Oldenburg, 2014; Vilarrasa and Rutqvist, 2017). We report the bottom hole pressure only if it is measured or calculated.

**2.2.6 General seismicity records**

General seismicity records include available information regarding the onset of seismicity (or its recording after a delayed installation of detecting networks), the seismicity lag time from the start of the operation, the number of events, and their occurrence depth range. The number of events comprises a brief text describing the number of all recorded events, not limited to the sequence of the maximum magnitude. We also document the *a*- and *b*-values of the Gutenberg-Richter empirical law,

which stands as a reference for statistical forecasting of the seismic hazard by explaining the magnitude-frequency distribution of earthquakes (Gutenberg and Richter, 1942):

$$logN = a - bM \tag{1}$$

where $N$ is the number of events with magnitudes equal to or larger than $M$. The *a*- and *b*-values denote the sequence productivity and the relative abundance of large- to small-magnitude events, respectively, and are extracted from earthquake

catalogs. For tectonic earthquakes, the *b*-value commonly approaches 1, meaning that events of magnitude $M \geq 2$ are statistically ten times more frequent than events of magnitude $M \geq 3$ for a given time window (Kanamori and Brodsky, 2004). The larger the *b*-value during induced seismicity, the larger the predominance of small earthquakes. From a physical point of view, high *b*-values may coincide with microseismicity and the opening of new fractures (tensile events) due to elevated overpressure close to the injection wellbore. Low *b*-values may denote reactivation (shear event) of pre-existing, critically

stressed faults, reflected in large stress drops and the corresponding moment magnitudes (Goertz-Allmann and Wiemer, 2013;



Zang et al., 2014). Nevertheless, establishing a physically-sound link between seismological observables and the geomechanical behavior of the subsurface remains a hot topic of active research.

The seismic hazard has reportedly been augmented with an increased tendency to induce larger events in the post-injection phase compared to co-injection seismicity for a number of high-profile induced earthquakes. These observations entail a

reduced $b$-value after wellbore shut-in. For instance, the $b$-value was observed to drop from 1.58 to 1.15 in Basel, Switzerland (Bachmann et al., 2011), deep geothermal project, and from 0.99 to 0.77 in Castor, Spain (Cesca et al., 2014), underground gas storage project between co- and post-injection seismicity. As a result, we report distinct parameter values for three different seismicity subsets: the background seismicity prior to the operation, seismic events during the injection phase, and seismicity trailing the wellbore shut-in.

**2.2.7 The maximum magnitude event**

In addition to general seismicity information, we record detailed information about the maximum magnitude event, including the possible depth range, the occurrence date, and the approximate distance from the injection borehole coming along with its magnitude and the magnitude field. We report moment magnitude $M_W$ whenever available; otherwise, we cite local magnitude $M_L$, and hardly any duration magnitude $M_D$ and body-wave magnitude $m_b$. If multiple magnitude types are available, we

preferentially include the moment magnitude in the cell and give the others in the comment. Converting the magnitude types is not straightforward and is left up to the end-users.

**2.2.8 Complementary remarks**

We designate a brief text to disclose the potential causal mechanism(s) of the induced earthquakes (see the Introduction Section) if resolved. Particularly, induced earthquakes are occasionally linked to multiple simultaneous anthropogenic

activities in the subsurface. For instance, regional-induced earthquakes in the Delaware Basin, Texas, are attributed to widespread shale fracking and wastewater disposal into deep and shallow aquifers (Zhai et al., 2021), giving rise to debates about the causative contribution of each activity. Insights obtained from analytical and numerical inspection of the temporal and spatial evolution of induced earthquake sequences and anthropogenic activities may help clarify such ambiguities. We also present any additional notes that help delineate the project, injection conditions, and observed seismicity. Specially, we

highlight features that may affect earthquake risk management, such as the aftershock sequence or the rupture nucleation beyond the target injection layer.

**2.2.9 Data sources**

The reported data in the database comes from publicly available resources, including scientific publications (books, peer-reviewed journal papers, or proceedings), relevant databases and published reports or dissertations. Accordingly, researchers

can refer to the references alphabetically cited in this part of the database to acquire further information. The references are



linked to an accompanying bibliography list provided at the database repository (see the Data availability Section). As the input parameters for the database come from various disciplines, multiple references are commonly used to complete data for each case. The reference(s) for each data entry is given as a comment on the corresponding cell in the Excel format.

**2.3 Data curation**

The vast majority of case examples of induced seismicity included in the database correspond to those compiled in the HiQuake database (Wilson et al., 2017), commonly referred to as a reference for human-induced earthquakes. Our primary auditing of the HiQuake database recognizes 551 case examples of injection-induced earthquakes, excluding those triggered during the development of conventional oil and gas resources essentially due to data scarcity. We find 349 of the cases inappropriate for inclusion in our database. A total of 320 discarded (micro)seismic sequences are associated with the hydraulic fracturing of shale gas plays in Oklahoma, for which the links with particular operating wells or injection programs are missing (Skoumal et al., 2018b). The majority of these hydraulic fracturing datasets (233 operations) suffer from a lack of the associated maximum earthquake magnitude data. Likewise, we discard an additional 22 fracturing-induced (micro)earthquake swarms in the Dawson-Septimus area, Canada (Roth et al., 2020) and 7 individual, sparsely located events attributed to geothermal operations, all missing similar basic seismic information. Collectively, we structure our database by considering 202 reported injection-induced earthquakes from HiQuake, complemented by five additional sequences: two recorded at the geologic $CO_2$ storage pilot sites of Heletz, Israel, and Hontomin, Spain, and three recorded during injection into cm-scale rock specimens in the laboratory.

We organize a comprehensive and systematic search for the variables described in Section 2.2. However, a handful of historical induced earthquakes lack rigorous characterization studies. For these events, many associated key parameters may be unavailable, with the corresponding cells in the database left blank. Therefore, we perform the second level of data auditing to exclude cases that lack basic information, such as injection data or host rock properties.

Data concerning fault properties are rare, primarily due to characterization limitations: (1) earthquakes are frequently induced on unmapped faults without prior characterization, (2) the reactivated fault is not necessarily crossed by any borehole and neither samples for laboratory studies nor wireline logging are available, and (3) sufficient and appropriately located field observations that enable in situ evaluation of the fault behavior, including microseismicity, pressure, and deformation measurements, scarcely exist. Consequently, the reported information in the database for many cases is limited to the faulting regime, strike, and dip inferred from the analysis of earthquake focal mechanisms. Nevertheless, we retain cases for which fault properties, in situ stress data, or some reservoir properties are unavailable to allow the users to benefit from the remaining reported parameters in special analyses. For instance, we keep cases for which the hydromechanical properties of the host rock are known, as this information is valuable for developing theoretical scaling relations between possible earthquake magnitudes and injection parameters.



Inputs for data fields can be either quantitative or qualitative and, thus, of either numeric, date, or text formats. Although data types may vary from one column to the other, all entries in each database column are necessarily of a unique type. Particularly,
we avoid entering any explanatory text into numeric fields but only integers or floats. Dates consistently conform to the ISO 8601 format, representing the year, followed by the month and the day, i.e., yyyy/mm/dd. Exact dates may be underreported in the public domain for some historical or even new cases of induced earthquakes, possibly due to inappropriate monitoring or recording. We replace missing dates on the month and day levels with the first month of the year and the first day of the month, respectively. If this is the case, explanatory comments are provided in the Excel file. Furthermore, we unify the database
by converting all numeric values to SI units. Accordingly, the database allows for direct calculations and data processing without requiring any unit and/or format conversion by other researchers.

We grade the entries for the host rock properties based on the data source: A for direct measurements of the parameter at the injection place, B for representative values of the same formation in an adjacent field or basin, and C for those rendering the typical behavior of the corresponding stratigraphy. We provide the grade of each entry in the comment for the corresponding
cell in the Excel file. We prioritize citing data in the order of reliability, i.e., grades A to C, narrowing down uncertainties of ensuing studies that users may build upon this database. The included grade-C values are published estimates of rock properties, commonly for numerical simulations, whose reliability in reproducing the rock behavior was adequately justified. If neither of the graded information is available, we avoid making independent and unverified assumptions to fill in the respective column for an event. Values resulting from grade-B information can also be accompanied by non-trivial levels of
uncertainty because the rock structure and its behavior may vary from place to place depending on the tectonic and environmental conditions undergone by the rock. In addition, different direct measurement techniques may give rise to discrepancies in the inferred parameter values. In cases where grade A is available, the values refer to laboratory evaluations unless otherwise stated in the comment. If measurements from multiple approaches are available, laboratory data are set as a reference for the sake of consistency with the remaining part of the database. The values derived from other sources are
provided in the comment, notwithstanding the notion that in situ field measurements may be more representative of the average rock mass behavior (Vilarrasa et al., 2013, Neuzil, 2019). Similar concerns could arise in parameter extraction from independent evaluations by different studies. For conflicting cases, we either merge the inferred data (ranges) or choose among them depending on the supremacy of the input data, techniques and assumptions applied for their assessment. The former strategy is commonly adopted for laboratory measurements of rock properties whereas the latter usually takes place for in situ
stress evaluations.

The configuration of predefined structure, styling, terminology, and data inspection criteria enables simultaneous data collection and curation. Nonetheless, we establish an ultimate integration phase to combine entries from individual contributors and provide a single unified representation of the database. Particularly, we scrutinize early grade-C data entries and whether they can be replaced with grade A or B from new insights obtained in the course of the database development. The database
could be updated in the future to add new data and/or modify the existing information if required.



## 3 Current database metrics

### 3.1 Earthquake distribution

We have collected so far data for 158 cases of injection-induced seismicity from 7 geologic gas storage projects (two natural gas and five carbon storage sites), 15 research projects, 54 tight and shale gas hydraulic fracturing, 58 deep geothermal programs and 24 wastewater disposal activities (summary in Table 1 and distribution map in Fig. 3). The numbers show that geothermal operations contribute the most to our induced seismicity database (36.5%), closely followed by hydraulic fracturing operations (34.6%), wastewater disposal (15.1%), research projects (9.4%) and underground gas storage (4.4%).

The database gathers data from 5 continents and 25 countries. However, neither the number nor the type of induced earthquakes is uniformly distributed around the globe. While the number of data entries is limited to one in African countries, the United States and Canada host 43 and 32 seismogenic injection operations, respectively. Furthermore, more than 58% of geothermal cases belong to European countries, 56% of hydraulic fracturing projects to Canada and 91% of wastewater disposal activities to the US. Nevertheless, the observed distribution patterns do not necessarily come up with a conclusive argument in favor of how seismogenic the conducted project types are in different countries; they could rather be attributed to the non-uniform distribution of geoenergy resources and development policies among different countries. The scatter in the recorded induced earthquakes are especially consistent with the overriding interest and growing investment of European countries in renewable energies, among them geothermal resources (Haas et al., 2011), and the prevalence of oil and gas resources in the US and Canada. Another key factor that could drive the observed trends is seismicity monitoring and reporting regulations that can vary among different project types and countries (Grigoli et al., 2017).

Interestingly, frequent hydraulic fracturing of shale gas reservoirs in the USA has experienced little to no publicized induced earthquakes (Verdon and Bommer, 2021), consistent with only 11 relevant cases reported in our database. We observe a similar lack of reported induced earthquakes for recent developments of shale gas plays in South America although natural earthquakes are prevalent in this region (Caruso, 2017). The paucity of fracturing-induced seismicity in these regions can be attributed to (1) detection limits, resulting mainly from inadequate installed monitoring networks, (2) underreporting when the seismicity is not deemed to pose safety risks and hazards to the local population and industrial operations, that is usually the case for regions of low population density (Wilson et al., 2017), (3) differences in operation and monitoring protocols and rules, and (4) systematic dissimilarities arising on regional and basin-wide bases from variations of the state of stress and characteristics of the stratigraphical setting. Skoumal et al. (2018a) argued that nearly aseismic stimulations of the Bakken and Marcellus shale plays in the Central and Eastern US stem from their distance to the seismogenic crystalline basement or the presence of isolating sediments diminishing hydraulic connections with the basement.



**Table 1. The number of collected induced earthquakes for each type of underground injection activity and their proportion in the database.**

| Injection operations | Number of cases in this database | Percentage of cases (%) |
|---|---|---|
| Hydraulic fracturing | 54 | 34.6 |
| Geologic gas storage | 7 | 4.4 |
| Geothermal energy | 58 | 36.5 |
| Research | 15 | 9.4 |
| Wastewater disposal | 24 | 15.1 |
| Total | 158 | 100 |


### 3.2 Earthquake magnitudes

The listed maximum earthquake magnitudes span a wide range from $M_W$ -7, detected in the laboratory during cm-scale fracture slip experiments (Goodfellow et al., 2015), to $M_L$ 6.6 at the Laugaland geothermal site, which is interestingly attributed to cold water reinjection at depths shallower than 1000 m (Flóvenz et al., 2015). An equally-sized large earthquake occurred in

connection with heat extraction at the Cerro Prieto geothermal field in Mexico (Glowacka and Nava, 1995). The database contains 36 $M > 4$ (~ 23%) events showing a meaningful high contribution by wastewater disposal projects. A total of 73 events (~ 46%) are considered with maximum magnitudes in the range between $M$ 2 and $M$ 4, which are dominantly linked with hydraulic fracturing of shale gas resources and geothermal field exploitation. The remaining cases, constituting a sizeable portion of the recorded earthquakes (counting to 30% and corresponding to 47 cases) have $M < 2$ (Fig. 4). The $M$ 2 is widely

adopted as a threshold below which the earthquakes may not be felt at the surface (Evans et al., 2012; Buijze et al., 2019). The vast majority of seismicity records of research projects and all CCS-induced seismicity cases belong to this magnitude range. It is very likely that many other fluid-injection projects would fall in this category with $M_{max} < 2$, but they have not received attention because the induced seismicity was not perceived by the local population and thus, are not included here or in other datasets.

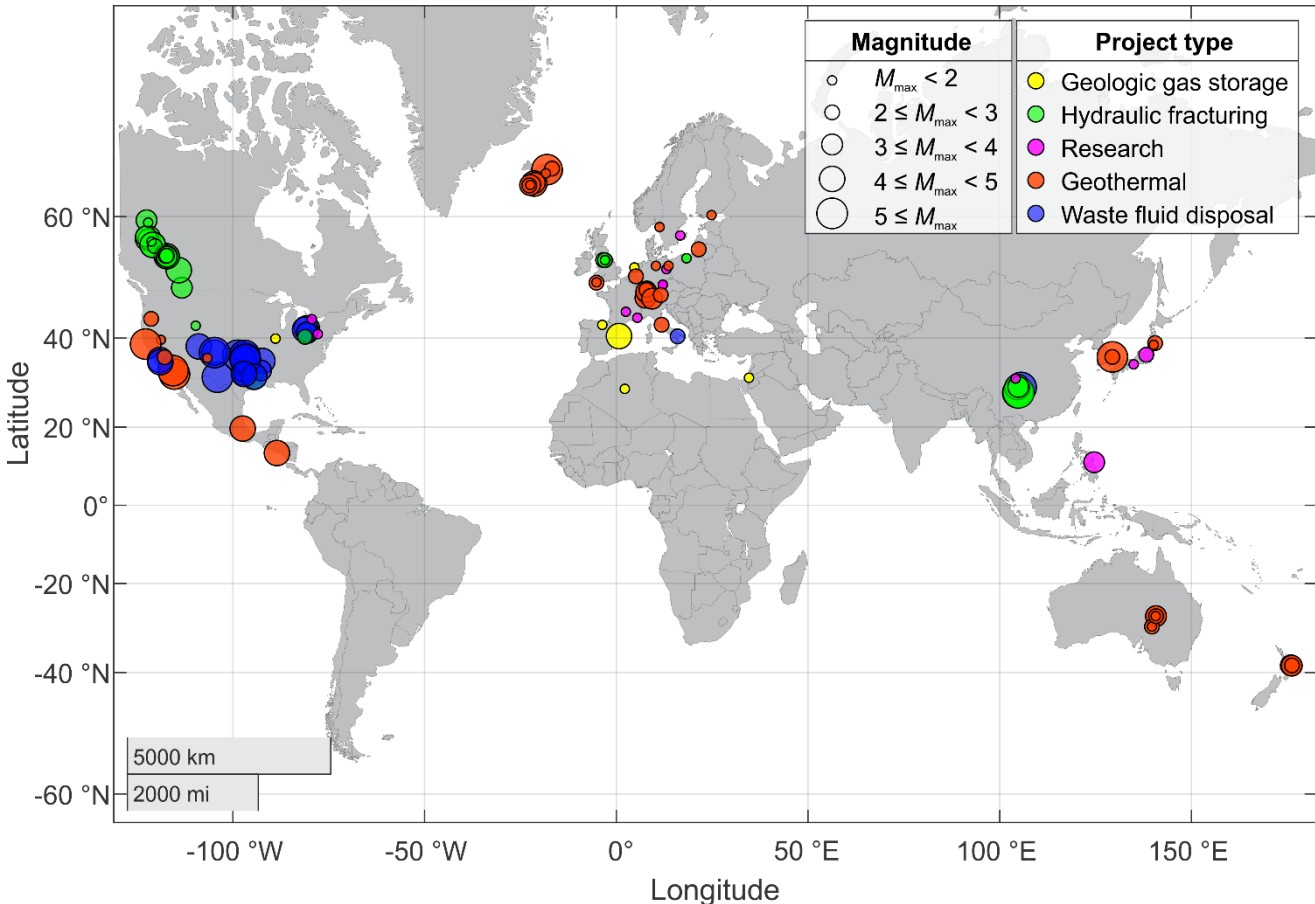

**Figure 3. The worldwide distribution of injection-induced seismicity cases included in the database.**

### 3.3 Geological setting

The included projects target 105 sedimentary rocks comprising carbonates, sandstones and shales, and 53 crystalline basement rocks. The crystalline basement is referred to here as all stratigraphic units underneath the sediments and is mainly composed of igneous (e.g., granite, basalt and diorite) and metamorphic rocks (gneiss and schist). Crystalline rocks serve as the main host to EGS projects. Only 13 earthquakes in the database, whose magnitudes are generally below $M$ 3, were induced in sedimentary geothermal reservoirs. In addition, some of these events are associated with injection or circulation in sediments lying directly over or proximal to the crystalline basement, where the existing faults are critically stressed and more seismogenic. Very prominent examples are (1) the $M_L$ 2.4 Unterhaching (Megies and Wassermann, 2014) and (2) the $M_L$ 3.5 Sankt Gallen (Diehl et al., 2017) in the Molasse Basin, Germany and Switzerland, respectively, and (3) the $M_L$ 2.4 Insheim and (4) the $M_L$ 2.7 Landau in the Upper Rhine Graben, Germany (Küperkoch et al., 2018). For these cases, a large share of



seismicity has demonstrably occurred in the crystalline basement. These observations are in agreement with multiple large-magnitude earthquakes that were nucleated on basement faults in Oklahoma and Texas, USA, as a result of wastewater disposal

in shallower aquifers (Verdon, 2014). Importantly, large-volume water injections into the Arbuckle group in close proximity to the basement are tightly linked with the $M_W$ 5.8 Pawnee, $M_W$ 5.7 Prague, $M_W$ 5.1 Fairview and $M_W$ 5.0 Cushing earthquakes. A variety of mechanisms, e.g., direct hydraulic connection, poroelastic stress perturbations and static stress transfer following fault slip, have proposed to govern alone or jointly the earthquake initiation in or rupture towards the crystalline basement (Johann et al., 2018; Vilarrasa et al., 2021, Zhai et al., 2021, Ge and Saar, 2022; Luu et al., 2022). Nevertheless, ample lines

of evidence from hydraulic fracturing of shales show that sedimentary rocks are prone to seismicity: the database contains cases that locate the majority of seismicity and the maximum magnitude event within the target sedimentary formation (e.g., the March 2019 $M_L$ 4.18 earthquake in Red Deer, Canada; Wang et al., 2020), in the overlying sediments (e.g., the January 2016 $M_W$ 4.1 earthquake in Fox Creek, Canada; Eyre et al., 2019) or underlying sediments (e.g., the December 2018 $M_L$ 5.7 and January 2019 $M_L$ 5.3 earthquakes in the Sichuan Basin, China; Lei et al., 2019), although along faults that may have their

root in the basement.

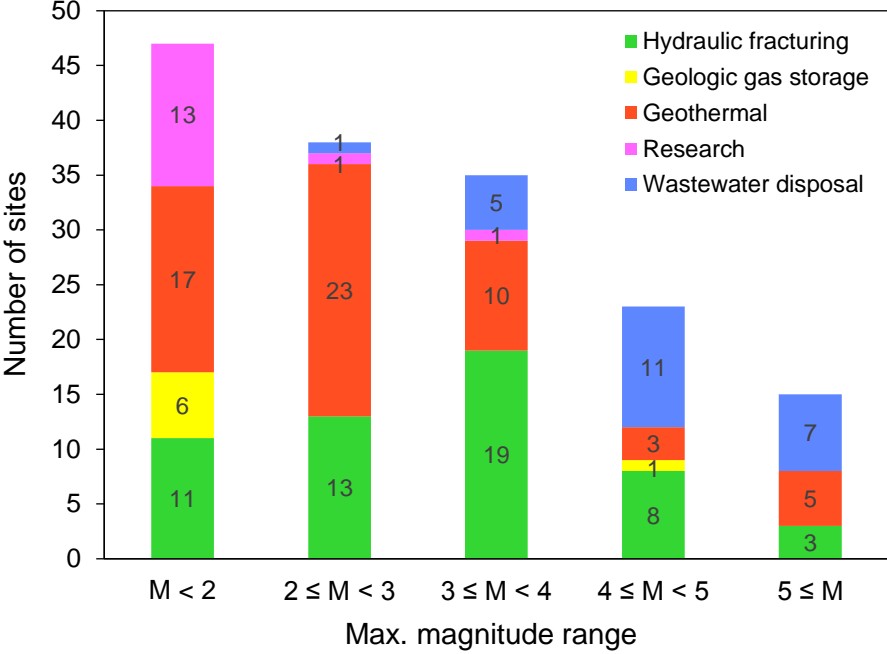

**Figure 4. Distribution of project types for different ranges of maximum earthquake magnitudes. Note that the data in this figure reflects the cases of reported induced seismicity. There are many other underground energy-related projects that are not included here. These cases would fall within the category of induced microseismicity that is not felt on the surface, i.e., M<2.**



## 3.4 Database statistics

Excluding data ranges by considering a representative average value for all parameters, as well as complementary remarks and references, the database so far comprises nearly 4,000 data entries (Table 2). In total, 36% of data entries ($n = 1,429$) belong to geothermal exploitation, 33% ($n = 1,297$) to hydraulic fracturing of shales, 15% ($n = 615$) to wastewater disposal, 10% ($n = 385$) to research projects and the remaining 6% ($n = 234$) to geologic gas storage. We observe almost identical distributions of database events and entered values across different operations (compare Table 1 and Table 2), implying that the considered induced earthquake categories are equally represented in the database. However, the total number of measurements may vary substantially from one variable to the other. For instance, formation names and stratigraphy information are known for the vast majority of the reported cases, enabling geological correlations between projects to help fill in data gaps for rock properties. Other extensively assessed parameters are hydraulic properties and stiffness of the host rock, in situ stress conditions, and fluid injection parameters (including total injected fluid volume and maximum injection rate and wellhead pressure). All these variables are key parameters for constraining the spatiotemporal evolution of seismicity. The database compiles this information for more than two-thirds of induced earthquakes. In contrast, fault properties have rarely been evaluated. Particularly, more than 85% of the datasets have no entries for fault thickness, permeability, and stiffness.

Key insights into the hydromechanical, operational, and seismic characteristics of geoenergy projects can be gleaned through a careful statistical inspection of the database (Fig. 5). Sediments targeted for geologic gas storage and wastewater disposal unsurprisingly feature high porosity and intrinsic permeability, which enable injection of significantly large fluid volumes at high rates and concurrently low wellhead pressures. The Bergermeer underground gas storage project in the Netherlands has by far the highest injection rate (463 m$^3$/s) and total injected fluid volume (4.3 billion m$^3$) among all collected cases. On the contrary, the majority of geothermal reservoirs, located in the deep crystalline basement, are characterized by low porosity and permeability. In these cases, earthquakes have frequently occurred during EGS stimulation during which the fluid injection pressures and rates are exceedingly high. However, peak injection volumes (>1 billion m$^3$) and rates (> 1 m$^3$/s) in geothermal projects are associated with long-term circulation operations in extensively fractured and highly permeable sedimentary, volcanic or metamorphic reservoirs (namely the Cerro Prieto, Mexico, and The Geysers and Salton Sea, USA, geothermal fields). It is worth stressing that these circulation systems may involve even no component of net injection volume depending on the simultaneous injection-extraction rates. Correspondingly, geothermal projects reflect a broad range of wellhead pressure, from 90 MPa achieved during hydraulic stimulation of a 6.1-km-deep reservoir in Helsinki, Finland, at one extreme down to near atmospheric injections under gravity drive into an underpressured reservoir of The Geysers at the other.



**Table 2. Number of collected data for different properties, presented in divisions defined by the project types.**

| Properties | Number of collected data | | | | | |
|---|---|---|---|---|---|---|
| | Hyd. fracturing | Geologic gas storage | Geothermal | Research | Wastewater disposal | Total |
| Formation name | 51 | 7 | 27 | 8 | 22 | 115 |
| Stratigraphy | 54 | 7 | 56 | 13 | 23 | 153 |
| Density | 40 | 2 | 32 | 3 | 3 | 80 |
| Porosity | 45 | 7 | 42 | 11 | 17 | 122 |
| Permeability | 44 | 7 | 40 | 10 | 18 | 119 |
| Young´s modulus | 43 | 7 | 35 | 9 | 12 | 106 |
| Poisson´s ratio | 45 | 5 | 35 | 9 | 11 | 105 |
| Biot coefficient | 21 | 3 | 11 | 3 | 3 | 41 |
| Friction angle | 5 | 2 | 9 | 2 | 3 | 21 |
| Cohesion | 4 | 1 | 10 | 2 | 3 | 20 |
| UCS | 10 | 0 | 15 | 4 | 2 | 31 |
| Tensile strength | 4 | 0 | 8 | 2 | 1 | 15 |
| Thermal Conductivity | 0 | 1 | 18 | 1 | 0 | 20 |
| Thermal expansion coefficient | 0 | 1 | 16 | 2 | 0 | 19 |
| Depth of basement | 22 | 3 | 11 | 0 | 5 | 41 |
| Stress regime | 29 | 7 | 47 | 6 | 19 | 108 |
| Overburden stress | 32 | 6 | 36 | 11 | 15 | 100 |
| Max. horizontal stress | 31 | 6 | 33 | 11 | 14 | 95 |
| Min. horizontal stress | 31 | 6 | 35 | 11 | 14 | 97 |
| Stress direction | 38 | 5 | 44 | 8 | 16 | 111 |
| Pore pressure | 35 | 6 | 29 | 7 | 13 | 90 |
| Temperature | 0 | 7 | 55 | 4 | 1 | 67 |
| Fault strike | 33 | 4 | 31 | 6 | 13 | 87 |
| Fault dip | 25 | 4 | 14 | 6 | 13 | 62 |
| Fault type | 26 | 4 | 12 | 6 | 14 | 62 |
| Thickness | 2 | 1 | 6 | 3 | 1 | 13 |
| Fault distance from injection | 4 | 4 | 6 | 4 | 3 | 21 |
| Intersection depth | 2 | 3 | 7 | 5 | 0 | 17 |
| Permeability | 4 | 2 | 12 | 5 | 2 | 25 |
| Normal Stiffness | 3 | 0 | 1 | 2 | 0 | 6 |
| Shear stiffness | 4 | 0 | 1 | 2 | 0 | 7 |
| Depth of injection | 46 | 7 | 56 | 12 | 23 | 144 |
| Injection start date | 43 | 7 | 56 | 12 | 23 | 141 |
| Fluid type | 25 | 7 | 20 | 14 | 9 | 75 |
| Injection temperature | 18 | 5 | 20 | 2 | 2 | 47 |
| Max. Injection rate | 36 | 7 | 49 | 15 | 21 | 128 |
| Injected volume | 30 | 7 | 32 | 15 | 21 | 105 |
| Net injection volume | 2 | 6 | 11 | 4 | 6 | 29 |
| Max. wellhead pressure | 33 | 5 | 39 | 13 | 15 | 105 |
| Max. bottomhole pressure | 3 | 4 | 3 | 2 | 1 | 13 |
| Seismicity onset | 34 | 4 | 22 | 5 | 18 | 83 |
| Seismicity lag time | 7 | 1 | 4 | 3 | 11 | 26 |
| Number of events | 34 | 7 | 43 | 12 | 19 | 115 |
| Depth of seismicity | 27 | 3 | 32 | 7 | 22 | 91 |
| G-R, during injection | 28 | 4 | 18 | 4 | 8 | 62 |
| $M_{max}$ | 54 | 6 | 58 | 15 | 24 | 157 |
| Depth of $M_{max}$ | 24 | 3 | 20 | 4 | 16 | 67 |
| Event distance from injection | 23 | 2 | 11 | 8 | 10 | 54 |
| Date of $M_{max}$ | 51 | 2 | 46 | 9 | 23 | 131 |
| Other parameters | 92 | 29 | 155 | 53 | 82 | 411 |
| Total | 1,297 | 234 | 1,429 | 385 | 615 | 3,960 |





Hydraulic fracturing of shale gas reservoirs shares a number of commonalities with EGS stimulations. These common features
mainly include the extremely low permeability of the reservoir rock and substantially high injection pressures and rates
required to induce and propagate hydraulic fractures. Furthermore, the target gas-bearing shales often possess comparable
stiffness (high Young´s modulus and low Poisson´s ratio) to those of geothermal reservoirs and sediments subjected to
wastewater injection. Corroborating this observation, high stiffness is inferred as a proxy for brittle shales, which are, by design
and intent, appropriate for creating and maintaining hydraulically-conductive fractures (Vafaie and Rahimzadeh Kivi, 2020).
Certainly, these brittle shales are different from clay-rich shales that stand on the lower bounds of Young´s modulus and upper
bounds of Poisson´s ratio variations for research projects and, likewise, the entire database. This low extreme of stiffness is
documented for Opalinus Clay at the Mont Terri URL, Switzerland, and an Upper Toarcian shale at the Tournemire URL,
France, which are widely considered representative caprocks for geologic $CO_2$ storage and hosts to nuclear waste disposal.
The rock stiffness range is bounded by a high Young´s modulus value of 95 GPa, measured for a gneiss rock in a deep injection
research project in Eastern Bavaria, Germany. Research projects broaden the registered ranges of injection rate and volume by
orders of magnitude down to $5 \times 10^{-10}$ $m^3$/s and $7 \times 10^{-9}$ $m^3$, respectively. These lower bounds of injection parameters were
drawn by fracture slip experiments on centimeter-scale specimens.

Eventually, variations of the *b*-value (indicative of the relative magnitude distribution of earthquakes) give a picture of the
discriminative features of earthquake sequences induced by different activities. We highlight two groups of seismicity:
seismicity with an average *b*-value of around 1, characterizing the recorded events during hydraulic fracturing and wastewater
disposal operations, and higher *b*-value seismicity (*b*-value > 1.4), observed during geothermal, geologic gas storage and
research projects. Conceptually, the former group is associated with the shear activation of major fault/fracture zones and a
higher probability of large-magnitude earthquakes, whereas the latter points to the dominance of spread and structureless
microseismic clouds (Zang et al., 2014).






**Figure 5. Boxplot for a number of database parameters. From bottom to top, the box indicates the first quartile, median and third quartile of the data. Whiskers represent the minimum and maximum values, excluding outliers. Outliers reside outside the range defined by 1.5 times the interquartile range added to the third quartile and subtracted from the first quartile**

**Data availability**

The .xls and .csv files of the database are available at the institutional repository Digital.CSIC: https://doi.org/10.20350/digitalCSIC/14813 (Kivi et al., 2022a). An associated list of references that were used to develop the database and a dictionary, including the definitions for all database parameters, are also provided in .docx format at the same address.



## Conclusions and perspectives

In this study, we have developed a comprehensive multi-physical database of injection-induced seismicity from various geoenergy applications: geothermal energy exploitation, shale gas development, underground gas storage, wastewater disposal,

and research projects. The database comprises a great variety of relevant properties, including general project information, rock properties, in situ site characteristics, fault attributes, operational parameters and recorded seismicity data. In the current release, nearly 4,000 data entries, covering 71 distinct variables for 158 projects (or project phases), are compiled from a critical review of more than 500 publications. Neither the frequency of earthquakes nor the type of triggering activities is uniformly distributed worldwide. The parameters span wide ranges of values, varying substantially among different project

types. We organize the database in simple flat-file formats to facilitate its utilization by researchers while keeping data directly readable by computer codes for implementation in model developments. All gathered data comply with a unique set of standards and quality requirements, ensuring high comparability, accuracy, and coherency of the data.

The high quality and large quantity and diversity of the collected data, integrating knowledge from geology, petrophysics, geomechanics and seismology, opens up opportunities for:

- improved assessment of the temporal and spatial occurrence of induced earthquakes;

    - recognizing the causative mechanisms of induced seismicity through direct data inspection or indirect inferences from physics-based numerical modeling, heavily depending on the provided data for parameterization and calibration;

    - highlighting possible relations between seismicity and operational parameters;

    - developing and validating empirical and/or theoretical scaling relations between the maximum earthquake magnitude

and injection parameters;

which collectively favor meaningful progress in forecasting induced seismicity hazards and proposing practical injection strategies to mitigate them. In addition, the collated data extend the opportunity to constrain modeling efforts, analytical or numerical, addressing other challenges in the safe and economical utilization of geological resources. Consequently, the database in a broader context contributes to unlocking the subsurface potential to accelerate achieving carbon neutrality.

Compiling data for a wide variety of parameters, plenty of induced seismicity events, such as those associated with shale gas fracturing in Oklahoma (Skoumal et al., 2018b), fail to fulfill the minimum requirements of being reported. Besides, the distribution of the existing data records is inhomogeneous, with frequently missing information for fault properties. This database is envisioned not to be static, but rather to be updated and extended by exploring newly published or potentially not-yet-considered data resources. We envisage potential improvements in data accessibility through the establishment of

collaborations with operators of geoenergy projects and authors of the existing compilations of relevant data fields, for example, physical and mechanical rock properties (P[3] database, Bär et al., 2020), fault properties (Scibek, 2020), in situ stress data (world stress map, Heidbach et al., 2018), and induced seismicity (Wilson et al., 2017). We also plan to create a publicly editable database interface on the GEoREST project website (www.georest.eu), through which we welcome contributions from all users to complement the database. Future improvements to the database include incorporating full induced seismicity

catalogs and detailed injection data (including time series of wellhead pressure and flow rate). This information allows for unveiling correlations in time and space between subsurface fluid injection and seismic activities. Following these extensions, the database would be organized in a mixed flat-file and rational structure to facilitate the desired data extraction and link with other rational databases, e.g., the P$^3$ database (Bär et al., 2020), using query-based languages (Gard et al., 2019).

**Author contribution**

VV, FP and IRK conceptualized the work. IRK, AB, HW, LW, SHH and VV collected the induced seismicity cases. IRK, FP and VV analyzed the data. IRK unified the database and wrote the paper with contributions from all authors.

**Competing interests**

The authors declare that they have no conflict of interest.

**Acknowledgments**

The authors acknowledge funding from the European Research Council (ERC) under the European Union's Horizon 2020 Research and Innovation Program through the Starting Grant GEoREST (www.georest.eu) under Grant agreement No. 801809. IRK also acknowledges support by the PCI2021-122077-2B project (www.easygeocarbon.com) funded by MCIN/AEI/10.13039/501100011033 and the European Union NextGenerationEU/PRTR. HW acknowledges the financial support received from the Secretariat for Universities and Research of the Ministry of Business and Knowledge of the 625 Government of Catalonia (AGAUR) and the European Social Fund (FI-2019). FP acknowledges funding from the European Union's Horizon 2020 Research and Innovation Programme through the Marie Skłodowska-Curie Action ARMISTICE (www.armistice-energy.eu) under grant agreement No. 882733. IDAEA-CSIC is a Centre of Excellence Servero Ochoa (Spanish Ministry of Science and Innovation, Grant CEX2018-000794-S funded by MCIN/AEI/ 10.13039/501100011033).

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
