# Peer review of "Global physics-based database of injection-induced seismicity"

_Earth System Science Data, 2022_

## Referee Comment (RC1)

**General comments**

This paper presents comprehensive physics-based database of injection-induced seismicity with descriptions related to its physical and statistical implications for understanding and forecasting of induced seismicity. I think the database adequately covers the vast range of physical parameters from worldwide injection-related geo-energy applications. The manuscript is well written and structured. I see the paper as a useful contribute to ESSD, if the below minor comments are handled appropriately.

**Specific comments**

1. This study seems not to discriminate the term "induced" and "triggered", although both terms indicate different nucleation mechanism in some studies (Dahm et al., 2015; McGarr et al., 2002, Ellsworth et al. 2019). Particularly, Ellsworth et al. (2019) classified the mainshock of the $M_w$ 5.5 Pohang earthquake as triggered seismicity, which was initiated by anthropogenic forcing and propagated beyond the bounds of the stimulated region. Other seismic events that occurred during hydraulic stimulations were termed as induced seismicity, of which magnitudes are limited to within the spatial dimension of the stimulated volume (Kim et al., 2022). As mentioned in the manuscript, $M_w$ 5.5 Pohang earthquake is regarded as a representative counter-example of magnitude scaling relations driven by McGarr (2014), but McGarr and Majer (2023) argue that the relationship is intended for earthquakes induced, not triggered. Thus, some descriptions regarding the term "induced" and "triggered" might be needed in the manuscript for the further understanding of the physical mechanism of seismicity.

2. In Figure 5, parameters with the logarithmic scale of y-axis such as permeability, maximum injection rate, maximum injection volume show that the mean value is plotted outside of the boxplot probably due to extremely large or small value of outliers. Particularly, the permeability for "research" is generally one or two order lower than the permeability for "geothermal", but mean values of both types indicate contrasting result. Mean values calculated by excluding the outliers can better represent the characteristics of the given parameters.

**Technical corrections**

1. L172. The project number is missing in first column of the given excel file.

2. Host rock properties in Figure 2. There is a typo in the unit of thermal expansion coefficient. (1/°K → 1/K)

3. L193: "hydrogeological" properties needs to be changed to "thermal and hydrogeological" properties, as thermal conductivity and thermal expansion coefficient are included in the reservoir rock properties.

**References**

Dahm T, Cesca S, Hainzl S, Braun T, Krüger F. Discrimination between induced, triggered, and natural earthquakes close to hydrocarbon reservoirs: a probabilistic approach based on the modeling of depletion-induced stress changes and seismological source parameters. J Geophys Res Solid Earth. 2015;120(4):2491–2509. https://doi.org/10.1002/2014JB011778.

McGarr A, Simpson D, Seeber L. Case Histories of Induced and Triggered Seismicity in International Handbook of Earthquake And Engineering Seismology. San Diego, CA: Acad. Press; 2002.

Ellsworth WL, Giardini D, Townend J, Ge S, Shimamoto T. Triggering of the Pohang, Korea, earthquake (Mw 5.5) by enhanced geothermal system stimulation. Seismol Res Lett. 2019;90(5):1844–1858. https://doi.org/10.1785/0220190102.

McGarr A. Maximum magnitude earthquakes induced by fluid injection. J Geophys Res Solid Earth. 2014;119(2):1008–1019. https://doi.org/10.1002/2013JB010597.

McGarr A, Majer EL. The 2017 Pohang, South Korea, Mw 5.4 main shock was either natural or triggered, but not induced. Geothermics. 2023;107. 102612.

---

## Author Comment (AC1)

Reply to "Comment on essd-2022-448" by Kwang-Il Kim (Referee)
Referee comment on "Global physics-based database of injection-induced seismicity" by Kivi et al., Earth Syst. Sci. Data, 2023

**General comments:**
This paper presents comprehensive physics-based database of injection-induced seismicity with descriptions related to its physical and statistical implications for understanding and forecasting of induced seismicity. I think the database adequately covers the vast range of physical parameters from worldwide injection-related geo-energy applications. The manuscript is well written and structured. I see the paper as a useful contribute to ESSD, if the below minor comments are handled appropriately.
*Authors' response: We thank the reviewer for his careful consideration and positive assessment of the paper.*

**Specific comments:**
1. This study seems not to discriminate the term "induced" and "triggered", although both terms indicate different nucleation mechanism in some studies (Dahm et al., 2015; McGarr et al., 2002, Ellsworth et al. 2019). Particularly, Ellsworth et al. (2019) classified the mainshock of the Mw 5.5 Pohang earthquake as triggered seismicity, which was initiated by anthropogenic forcing and propagated beyond the bounds of the stimulated region. Other seismic events that occurred during hydraulic stimulations were termed as induced seismicity, of which magnitudes are limited to within the spatial dimension of the stimulated volume (Kim et al., 2022). As mentioned in the manuscript, Mw 5.5 Pohang earthquake is regarded as a representative counter-example of magnitude scaling relations driven by McGarr (2014), but McGarr and Majer (2023) argue that the relationship is intended for earthquakes induced, not triggered. Thus, some descriptions regarding the term "induced" and "triggered" might be needed in the manuscript for the further understanding of the physical mechanism of seismicity.
*Authors' response: We generally agree that defining the terms "induced" and "triggered" can be of help in some studies to highlight particular features of earthquakes, e.g., their origin or causing mechanisms. Yet, the decision on whether an earthquake has been triggered or induced is usually debatable and a consensus is not always reached. Indeed, there is no quantitative threshold to discriminate between the two types of seismicities. For this reason, we have preferred not to attempt to distinguish in our database between induced and triggered seismicity. In the manuscript, we tend to consistently use the term induced for all earthquakes of anthropogenic origin. Yet, for the case of Pohang earthquake, we already mentioned its triggered origin: "…the 2017 $M_{max}$ 5.5 Pohang earthquake in Korea, triggered by stimulation of an EGS,…" (page 4, lines 89-90). In the revised version of the manuscript, we will add a statement to emphasize that we do not distinguish between induced and triggered earthquakes and consistently use the term induced for all earthquakes of anthropogenic origin, with the exception of Pohang, which has been studied in detail and a committee of experts agreed on its triggered origin.*
*Regarding the existing magnitude scaling relations, we already mentioned their limitations. We will emphasize that the seismic forecasting capability of these models is limited. We believe that the compiled database will help testing the applicability of these models and develop alternatives that better include the underlying physics.*

2. In Figure 5, parameters with the logarithmic scale of y-axis such as permeability, maximum injection rate, maximum injection volume show that the mean value is plotted outside of the boxplot probably due to extremely large or small value of outliers. Particularly, the permeability for "research" is generally one or two order lower than the permeability for "geothermal", but mean values of both types indicate contrasting result. Mean values calculated by excluding the outliers can better represent the characteristics of the given parameters.

*Authors' response: We thank the reviewer for this comment. The average values have been calculated after exluding outliers and will be used in the revised manuscript. The new plot is shown below.*

[Figure]

**Figure 5.** Boxplot for a number of database parameters. From bottom to top, the box indicates the first quartile, median and third quartile of the data. Whiskers represent the minimum and maximum values, excluding outliers. The mean values are also calculated after excluding outliers. Outliers reside outside the range defined by 1.5 times the interquartile range added to the third quartile and subtracted from the first quartile

**Technical corrections**

1. L172. The project number is missing in first column of the given excel file.

2. Host rock properties in Figure 2. There is a typo in the unit of thermal expansion coefficient. (1/°K → 1/K)

3. L193: "hydrogeological" properties needs to be changed to "thermal and hydrogeological" properties, as thermal conductivity and thermal expansion coefficient are included in the reservoir rock properties.
*Authors' response: Corrections have been made in the revised manuscript.*

---

## Author Response (AR1)

**Response to reviewers comments on the paper "essd-2022-448"**

We discuss below the comments made by two reviewers, our response to them and how we address them in the manuscript. To facilitate reading, the original comments are provided in a standard font, our responses in italics blue font, and a summary of the changes made in the text in italics brown font. Page and line numbers of the manuscript with track changes are used to address changes made in the text.

**Response to comments by Dr. Kwang Il Kim**

General comments:
This paper presents comprehensive physics-based database of injection-induced seismicity with descriptions related to its physical and statistical implications for understanding and forecasting of induced seismicity. I think the database adequately covers the vast range of physical parameters from worldwide injection-related geo-energy applications. The manuscript is well written and structured. I see the paper as a useful contribute to ESSD, if the below minor comments are handled appropriately.
*Authors' response: We thank the reviewer for his careful consideration and positive assessment of the paper. We have tried to carefully address the reviewer´s comments in the paper.*

Specific comments:
1. This study seems not to discriminate the term "induced" and "triggered", although both terms indicate different nucleation mechanism in some studies (Dahm et al., 2015; McGarr et al., 2002, Ellsworth et al. 2019). Particularly, Ellsworth et al. (2019) classified the mainshock of the Mw 5.5 Pohang earthquake as triggered seismicity, which was initiated by anthropogenic forcing and propagated beyond the bounds of the stimulated region. Other seismic events that occurred during hydraulic stimulations were termed as induced seismicity, of which magnitudes are limited to within the spatial dimension of the stimulated volume (Kim et al., 2022). As mentioned in the manuscript, Mw 5.5 Pohang earthquake is regarded as a representative counter-example of magnitude scaling relations driven by McGarr (2014), but McGarr and Majer (2023) argue that the relationship is intended for earthquakes induced, not triggered. Thus, some descriptions regarding the term "induced" and "triggered" might be needed in the manuscript for the further understanding of the physical mechanism of seismicity.
*Authors' response: We generally agree that defining the terms "induced" and "triggered" can be of help in some studies to highlight particular features of earthquakes, e.g., their origin or causing mechanisms. Yet, the decision on whether an earthquake has been triggered or induced is usually debatable and a consensus is not always reached. Indeed, there is no quantitative threshold to discriminate between the two types of seismicity. For this reason, we have preferred not to attempt to distinguish in our database between induced and triggered seismicity. In the manuscript, we tend to consistently use the term induced for all earthquakes of anthropogenic origin. Yet, for the case of Pohang earthquake, we already mentioned its triggered origin: "...the 2017 $M_{max}$ 5.5 Pohang earthquake in Korea, triggered by stimulation of an EGS,... " (page 4, lines 89-90). In the revised version of the manuscript, we have emphasized that we do not distinguish between induced and triggered earthquakes and consistently use the term induced for all*

*earthquakes of anthropogenic origin, with the exception of Pohang, which has been studied in detail and a committee of experts agreed on its triggered origin.*

*Regarding the existing magnitude scaling relationships, we already mentioned their limitations. For example, regarding McGarr's magnitude scaling relationship with the cumulative fluid injection volume, no universal quantitative threshold exists to objectively distinguish between induced and triggered earthquakes. Furthermore, even if this model worked well for induced earthquakes, we could not know a priori the induced or triggered essence of an earthquake before it happens. Accordingly, we have emphasized in the revised manuscript that the seismic forecasting capability of these models is limited. We have also mentioned that the compiled database will help test the applicability of these models and develop alternatives that better include the underlying physics.*

**Changes in the manuscript:**
*Clarifications have been made in the manuscript:*
*Page 5, lines 119-123: "It should be noted that the terms induced and triggered are occasionally employed in the literature to discriminate between man-made earthquakes depending on their origin or causing mechanisms (McGarr et al., 2002; Ellsworth et al., 2019; Buijze et al., 2020). We, however, do not distinguish between induced and triggered earthquakes hereafter in the article and in the database and consistently use the term induced for all earthquakes of anthropogenic origin".*
*Page 4, line 90: "However, caution should be taken when employing scaling relations of induced seismicity as their seismic forecasting capability is limited".*

2. In Figure 5, parameters with the logarithmic scale of y-axis such as permeability, maximum injection rate, maximum injection volume show that the mean value is plotted outside of the boxplot probably due to extremely large or small value of outliers. Particularly, the permeability for "research" is generally one or two order lower than the permeability for "geothermal", but mean values of both types indicate contrasting result. Mean values calculated by excluding the outliers can better represent the characteristics of the given parameters.

*Authors' response: We thank the reviewer for this comment. The average values have been calculated after excluding outliers and have been used in the revised manuscript. The new plot is shown below.*

[Figure]

**Figure 5.** Boxplot for a number of database parameters. From bottom to top, the box indicates the first quartile, median and third quartile of the data. Whiskers represent the minimum and maximum values, excluding outliers. The mean values are also calculated after excluding outliers. Outliers reside outside the range defined by 1.5 times the interquartile range added to the third quartile and subtracted from the first quartile

Technical corrections

1. L172. The project number is missing in first column of the given excel file.
*Authors' response: Project numbers have been added to the first column of the excel file.*

2. Host rock properties in Figure 2. There is a typo in the unit of thermal expansion coefficient. (1/°K → 1/K)
*Authors' response: The typo in the figure has been corrected.*

3. L193: "hydrogeological" properties needs to be changed to "thermal and hydrogeological" properties, as thermal conductivity and thermal expansion coefficient are included in the reservoir rock properties.
*Authors' response: This change has been made in the revised manuscript.*

**Response to comments by Prof. Gillian R. Foulger**

The paper by Kivi et al. introduces a global physics-based database of injection-induce seismicity. I recommend acceptance with the option of making the following very small number of minor tweaks.

The authors provide a publicly available database which is very positive and will without doubt be welcomed by diverse stakeholders and researchers. Providing information to fill in critical gaps in the HiQuake database for well-documented cases is a very positive step.

The early part of the paper provides a good concise review of the subject. The paper continues on to explain the rationale of the database and details of its contents. The work was clearly thought through very carefully and reasons for decisions are given.

The paper is authoritatively written and the relevant descriptions of data categories thorough. The database accords with official international standards. The illustrations and tables are elegant and clear.

The English is excellent throughout the main body of the paper.

*Authors' response: We thank the reviewer for her careful consideration and positive assessment of the paper. We have tried to carefully address the reviewer´s comments in the paper.*

I noticed a small number of awkward/wrong expressions in the Abstract that would benefit from a light editing: 1) "if intensely shaking the ground" should be reworded, 2) "these data are hardly gathered" would be better "these data are challenging to gather", 3) "data aims" should be "data aim", 4) "Conclusively" could be deleted.

*Authors' response: The suggested corrections have been implemented in the Abstract.*

Line 125: It is not clear what is meant by "We rely on HiQuake in the consensus that anthropogenic activities induced all reported earthquakes." Please rephrase this. HiQuake lists all earthquakes proposed, on the basis of scientific arguments, to have been human-induced. Simple inclusion in HiQuake says nothing about the likelihood that the proposals are true, and cases vary from highly unlikely to virtually certain to be human induced. It would be very helpful if this critical but widely unappreciated feature of HiQuake could be made clear here in the text.

*Authors' response: Following the reviewer's comment, we have modified our explanation of the natural or induced origin of the earthquakes included in the database.*

*Changes in the manuscript:*
*Page 6, lines 128-134: We have deleted the sentence "We rely on HiQuake in the consensus that anthropogenic activities induced all reported earthquakes". We have added "We build our database mainly upon HiQuake, the holistic and invaluable compilation of earthquakes proposed, on scientific grounds, to be induced by human activities. However, Wilson et al. (2017) and Foulger et al. (2018) point to varying degrees of certainty, from strongly unlikely to virtually certain, that the reported earthquakes in HiQuake have been anthropogenically induced".*

Line 186: Foulger et al. (2018) mention three CCS projects that are thought to be seismogenic.

*Authors' response: We have reworded the sentence for clarification. We have already mentioned in the first paragraph of Section 3.2 the three seismogenic CCS projects reported in HiQuake.*

*Changes in the manuscript:*
*Page 9, line 196: "Megatonne geologic carbon storage projects have been accompanied by low seismic activity" instead of "Arguably, megatonne geologic carbon storage projects have remained seismically quiescent".*

Section 2.2.6: It would be useful to cite errors in the *b*-values mentioned as the errors in *b* are frequently larger than the differences between the *b*-values under discussion. It is thus common to read interpretations of variations in *b* that are not statistically significant. The text at line 548 onwards also needs attention in this respect.
*Authors' response: We agree with the reviewer and have adapted the text accordingly.*

*Changes in the manuscript:*
*Corrections have been made to the manuscript to highlight the point raised by the reviewer:*
*Page 14, lines 341-343: "For instance, the b-value has shown meaningful reductions from 1.58±0.05 to 1.15±0.07 in Basel, Switzerland (Bachmann et al., 2011), deep geothermal project, and from 2.0±0.3 to 1.1±0.1 in Castor, Spain (Ruiz-Barajas et al., 2017), underground gas storage project between co- and post-injection seismicity".*
*Page 23, lines 567-570: "Nevertheless, evaluations of the b-value are commonly accompanied by large errors that challenge the treatment of the variations in the b-value as statistically meaningful. Hence, conclusive statements on the differences between induced seismicity patterns solely based on this parameter should be avoided (Shi and Bolt, 1982)".*